# Elucidating the Mechanism of Coumarin Homodimerization Using 3-Acetylcoumarin Derivatives

**DOI:** 10.3390/molecules30030651

**Published:** 2025-02-01

**Authors:** Kristina B. Simeonova, Ana I. Koleva, Nevena I. Petkova-Yankova, Anna-Mariya R. Zlatanova, Vesela Lozanova, Rositca D. Nikolova, Petko St. Petkov

**Affiliations:** 1Faculty of Chemistry and Pharmacy, Sofia University “St. Kliment Ohridski”, 1164 Sofia, Bulgaria; ohks@chem.uni-sofia.bg (K.B.S.); ohak@chem.uni-sofia.bg (A.I.K.); nipetkova@chem.uni-sofia.bg (N.I.P.-Y.); an.zlatanova@gmail.com (A.-M.R.Z.); 2Department of Medicinal Chemistry and Biochemistry, Medicinal University of Sofia, 1431 Sofia, Bulgaria; vlozanova@medfac.mu-sofia.bg

**Keywords:** reaction mechanism, homodimerization, 3-acetylcoumarin, density functional theory calculations, coumarin, DFT modeling, 2-oxo-2H-1-benzopyran

## Abstract

The current study is a continuation of our previous investigations into the radical homodimeric reaction mechanism of 3-acetylcoumarin. In the current study, the effects of different substituents on the coumarin ring of 3-acetylcoumarin are investigated both experimentally and theoretically. Several 3-acetylcoumarin derivatives (substituted at C-6, C-7, and C-8) were tested in the optimized reaction conditions under ultrasound irradiation, and biscoumarin species were isolated and characterized. The elucidation of the substituent’s effect was further investigated by means of DFT calculations (free-energy calculations, NBO analysis), both in the initial substituted coumarins and in the formed radicals. It was observed that the presence of substituents at the C-6 and C-8 positions in the coumarin moiety would not affect significantly the formation of a radical, while a group at position C-7 could either stabilize or destabilize the formed radical depending on the electronic properties of the substituent.

## 1. Introduction

During the last several decades, the synthesis of coumarin representatives has been of great interest [1,2,3,4,5,6,7,8] due to the vast usage and application of the coumarin moiety as a building block in different systems with therapeutic potential [9,10,11,12,13]. Furthermore, a variety of coumarin compounds have exhibited antitumor and antiproliferative effects on several tumor cell types [14,15,16,17] and are successfully used as photodynamic and photochemotherapeutic agents [18] and biomarkers [19,20]. Other significant pharmacological properties, such as anti-Alzheimer [20,21], anti-HIV [5,21,22], antifungal [20], antiviral [20], antibacterial [23,24], anti-inflammatory [24,25], antioxidant [20,26], and anticoagulant [27] activities, were as well studied.

Due to the importance of this class of compounds, it is of great interest to be able to predict the reaction outcome.

Based on our previous work [28,29] and the desire to better understand the reaction pathways of dimerization, we continued our investigations into the formation of biscoumarins under different conditions by using both synthetic and computational modeling experiments. The main aim of this research was to elucidate the effect of the substituents on the rate of the reaction and to test the possibility of altering the mechanism, if possible, by changing the electron density in the coumarin moiety by applying different groups in the benzene ring.

## 2. Results and Discussion

In our previous studies [28,29] on 3-acetylcoumarin’s **1a** homodimerization, we hypothesized that the dimerization process occurs through the formation of radical species initiated by the cavitation effect of ultrasound irradiation.

Two approaches for the synthesis of **2a** were accomplished, Figure 1. The first approach included metal zinc and chloroacetic anhydride in THF/Et_2_O media (Method A, Figure 1) [28] under sonication conditions (20 kHz, 40 °C). The starting material was fully converted within 10 min, and the biscoumarin **2a** was isolated with a 92% yield. These results prompted us to further test the ability to modify the reaction condition by using less-toxic alternatives. Therefore, in the second method, we successfully replaced the chloroacetic anhydride with different metal salts. The best results were obtained when Zn (OAc)_2_ (Method B, Figure 1) was used [29]. The reaction was again initiated by ultrasound irradiation (37 kHz, 40 °C, and 30% wave amplitude). After optimizing the conditions, the full conversion of the acetylcoumarin **1a** was observed within 10–15 min with a 90% yield. Theoretical calculations on the formation of the biscoumarin species and the intermediates in the reaction were performed. The results suggested that the radical mechanism is more favorable when dimers are formed [29].

Therefore, to continue our thorough investigation and to better understand the mechanism of the reaction, it is of great importance to elucidate the effect of the substituent in the benzene ring in 3-acetylcoumarin **1a** on the reaction outcome (reaction time and yields). Hence, a series of acetyl derivatives bearing electron-donating groups in different positions and halogens were selected, Table 1.

As can be seen from the results presented in Table 1, the conditions that favor the formation of bisdihydroacetyl coumarins **2b–2f** with better yields were when chloroacetic anhydride and zinc under ultrasound irradiation were used. An exception was observed when coumarin **1g** was tested in the reaction. The obtained yields are significantly lower compared to the biscoumarins **2b–2f**. This might be due to steric hindrance of the reaction center or in the formed product.

It is interesting to note that, even though the yields from the homodimerization reaction obtained by Method B were low, the reaction time for the full conversion of the starting coumarin was shorter. The lower yields might be a result of undergoing a side reaction—additional products were observed while following the reaction by TLC. The additional products might be due to the slightly basic conditions when Zn(OAc)_2_x2H_2_O was used. It is known from the literature [7,30,31,32] that coumarins are not very stable in basic conditions because the lactone moiety could undergo a ring opening. In both cases, the reaction with the acetyl derivative bearing a -OMe group at position 7 was not successfully accomplished. This result might be explained by the instability of the formed radical species 2c-I-3, where the same C atom would bear the radical and the methoxy group, Figure 2.

Due to this observation, an experiment with 7-methyl-3-acetylcoumarin **1h** was performed. The reaction was carried out in the presence of chloroacetic anhydride and Zn and initiated by ultrasound irradiation. The starting coumarin **1h**, Method A, converted to the biscoumarin species **2h** for 120 min with 82% yield. The reaction needed additional heating of 45 °C for accomplishing in comparison with the other derivatives where the bath was heated to 40 °C. Compound **1c** did not react even when the temperature was higher.

Therefore, to clarify the proposed radical mechanism of the 3-acetylcoumarin derivatives to form the homodimeric product, quantum chemical calculations based on DFT were performed. Eight derivatives of 3-acetylcoumarin containing -OMe, -Cl, -NO_2_, -Et_2_N, and -Me functional groups were investigated. The calculated free energies for the formation of the corresponding radicals show that the radical formation is slightly endergonic, similar to the formation of the radical in the unsubstituted 3-acetylcoumarin **1a**. What can be seen from the calculations, though, is the effect the substituents have on the calculated free energies, depending on their position and electronic effects. If a substituent is added in C-6 or C-8 position, the calculated relative free energies do not change significantly compared with the unsubstituted 3-acetylcoumarin radical (ΔG = +54 kJ/mol for 3-acetylcoumarin-radical, entry 1; ΔG = +55 kJ/mol for 6-Cl-radical, entry 2; ΔG = +59 kJ/mol for 6-OMe-radical, entry 3; ΔG = +48 kJ/mol for 6-NO_2_-radical, entry 4; ΔG = +62 kJ/mol for 8-OMe-radical, entry 9), Table 2.

In positions C-6 or C-8, the calculated free energies for the formation of the radicals vary only by 14 kJ/mol, regardless of the electronic effects of the substituents. The reason for this could be that the electron density of the unpaired electron, localized in position C-4 and where the homodimerization occurs, could not conjugate with the substituents in the C-6 and C-8 positions. Moving towards the 7-substituted 3-acetylcoumarin derivatives (entries 5–8, Table 2), some differences in the calculated free energies could be observed, relative to the unsubstituted 3-acetylcoumarin. If an electron-donating group is present in position C-7, the calculated free energies increase by 12 to 39 kJ/mol (ΔG = +54 kJ/mol for 3-acetylcoumarin-radical, entry 1; ΔG = +70 kJ/mol for 7-methoxy-radical, entry 5; ΔG = +93 kJ/mol for 7-diethylamino-radical, entry 7; ΔG = +66 kJ/mol for 7-methyl-radical, entry 8, Table 2), making the formation of the radical species more unfavorable. This could be due to the reason that EDGs (electron-donating groups) promote electron density towards the formed radical in position C-4. On the other hand, if an electron-withdrawing group is added in position C-7, the formation of the radical is more favorable (ΔG = +54 kJ/mol for 3-acetylcoumarin-radical, entry 1; ΔG = +19 kJ/mol for 7-nitro-radical, entry 6, Table 2), which is due to the ability of the NO_2_ group to withdraw electron density from position C-4, thereby delocalizing the electron density of the unpaired electron in position C-4 over the other atoms in the coumarin system. This conclusion is supported by the plots of the spin densities of the investigated radicals of 3-acetylcoumarin and its derivatives using an isovalue of 0.005. The presented spin densities additionally indicate that the unpaired electrons are located mainly on the Zn atom, coordinated towards the O atoms, from the lactone ring and in positions C-4, C-5, and C-7, Figure 1. Moreover, in positions C-6 and C-8, there is no spin density, which further proves the fact that substituents in those positions do not affect the reaction and the formation of the desired homodimeric product directly.

To analyze the electron density change upon radical formation, NBO analysis for the initial compounds (3-acetylcoumarin and its derivatives) and the formed radicals, Table 3, was performed.

The focus was on the change in the charge distribution on the C-4 atom, where the homodimerization process occurs via the formation of a new C-C covalent bond between the two monomers. With this analysis, the effect of the substituents on the charge distribution after the formation of the radicals, and more specifically the effect of the substituents in the C-7 position of the 3-acetylcoumarin moiety, as they have a direct effect on position C-4, was highlighted. The most notable difference that could be observed is in the 7-methoxy, 7-diethylamino-, and 7-nitro- derivatives of the 3-acetylcoumarin, where the difference in the C-4 charge is much more prominent compared to the other formed radicals (Δq = −0.152 for the 7-methoxy-radical, entry 5 and Δq = −0.167 for the 7-diethylamino-radical, entry 7, compared to Δq = −0.143 for the 3-acetylcoumarin-radical, entry 1, Table 3). This is due to the electron-donating properties of the methoxy- and diethylamino- groups, where a conjugation between the substituent at C-7 with the C-4 atom could occur. As we know from the experimental data, a product containing a 7-methoxy- substituent could not be formed. We could assume that the EDGs destabilize the radical, Figure 2. Even though the methyl- substituent in position C-7 also has weak electron-donating properties, the changes in the charge distribution are not as prominent compared to the other EDG in position C-7 (Δq = −0.147 for 7-methyl-radical, entry 8, Table 3). If we focus on the 7-nitro-radical, we can see the slightest charge variation before and after the formation of the radical (Δq = −0.033 for 7-nitro-radical, entry 6, Table 3). As it was discussed earlier, such a small difference in the charge distribution on atom C-4 is due to the electron-withdrawing effect of the NO_2_ group, as well as the possibility of substituents in position C-7 to conjugate with the C-4 atom.

To further understand the local reactivities of the 3-acetylcoumarin and its derivatives, we have calculated and analyzed the Fukui indices, Table 4. To analyze the radical formation, we focus on the ƒ^0^ values, as they represent the average of ƒ^+^ (electrophilic susceptibility) and ƒ^−^ (nucleophilic susceptibility). A higher ƒ^0^ value suggests that the corresponding molecular site is more prone to radical formation, whereas lower values indicate reduced radical reactivity. The key observations from the calculated ƒ^0^ values indicate that (1) 8-OMe (ƒ^0^ = 0.088) shows the strongest tendency for radical formation. The strong electron-donating MeO-group at this position destabilizes the radical, making the C-4 atom highly reactive. (2) Substituents like 7-Me (ƒ^0^ = 0.076) and 6-OMe (ƒ^0^ = 0.069) also enhance radical formation, although not as strongly as 8-OMe. (3) The 6-Cl (ƒ^0^ = 0.064) slightly increases the radical susceptibility compared to the parent coumarin H- (ƒ^0^ = 0.056), and (4) 6-NO_2_ (ƒ^0^ = 0.043) and 7-OMe (ƒ^0^ = 0.049) lower the tendency for radical formation at C-4, Table 4. This is due to the electron-withdrawing effects of -NO_2_ and the stabilizing resonance effect of the MeO-group.

The calculated Fukui indices, as indexes of local reactivity, are in line with the experimental trend, where the 7-OMe derivative of the 3-acetyl coumarin did not react in the presented conditions for the formation of the desired biscoumarin product, while the 8-OMe derivative, with the strongest tendency for radical formation, produces biscoumarin with the highest yield for short time.

The atomic charges and Fukui indices are the local indexes of reactivity, while the frontier orbital energies, such as singly occupied molecular orbitals (SOMO), are a global index of reactivity. For this reason, we considered SOMO energy change upon radical formation. The change in the energies of the SOMO of the investigated radicals with the same substituent on different positions was investigated. Here, a similar trend to the NBO charge variations could be observed. The electron-donating groups (EDG), such as the methoxy-group (-OMe) in positions C-6 and C-8, show higher SOMO energy, i.e., it destabilizes with respect to the C-7 position, which indicates the more efficient EDG effect of the -OMe group in C-7 position. Such destabilization of the SOMO is also in good agreement with the higher free energy (11 kJ/mol) for the formation of a radical with the -OMe group at C-7. On the other hand, EWGs (electron-withdrawing groups), such as NO_2_, destabilize SOMO in C-7, with respect to C-6, thus leading to easier radical formation with the -NO_2_ group in C-7 by 30 kJ/mol.

We have investigated the changes in the core level (1s) energies of the C-4 as an indirect measure of chemical reactivity by reflecting changes in the electronic environment around the active site, Table 5.

We focused our attention on the core-level shifts of C-4, before and after radical formation with the -OMe and -NO_2_ substituent in position C-6 and position C-7, respectively, with respect to the 3-acetylcoumarin. The presence of the NO_2_ group in position C-6 does not play a role in the C-4 core level shifts upon radical formation (Δ = 1.622 eV for the 6-NO_2_-radical, entry 4, and Δ = 1.641 eV for the unsubstituted 3-acetylcoumarin radical, entry 1, Table 5). The same trend could be observed in the presence of the OMe group in position C-6, where the change in the shift is even slighter (Δ = 1.616 eV for the 6-OMe-radical, entry 3, and Δ = 1.641 eV for the unsubstituted 3-acetylcoumarin radicals, entry 1, Table 5). This indicates, again, that substituents in position C-6 of the 3-acetylcoumarin have a negligible electronic effect on the C-4 atom. If we consider the substituents in position C-7, a slight increase in the shift could be seen for the C-4 core level if the OMe group is presented (Δ = 1.656 eV for 7-OMe-radical, entry 5, Table 5) and a decrease in the shift in the presence of NO_2_ group (Δ = 0.957 eV for 7-NO_2_-radical, entry 6, Table 5), compared to the unsubstituted 3-acetylcoumarin radical. The most significant change overall in the shifts of the core levels of the C-4 atom could be observed in the presence of the EWG in position C-7, for instance, the NO_2_ group, thus confirming the impact of the electron-withdrawing group in this position on the coumarin moiety.

Moving towards the product of homodimerization, we have considered a few different metal salts that could be formed during the reaction, considering the experimental conditions. In our previous study, it was noted that homodimerization occurs easily with both ZnCl_2_ and Zn(OAc)_2_ salts. The outcome of the reaction was comparable. However, the workup of the reaction was complex due to the solubility of the ZnCl_2_. Therefore, the experiments were further performed with Zn(OAc)_2_ [29]. In order to report consistent data with our previous studies, we have taken into consideration the use of ZnCl_2_ as a metal salt, as well as chloroacetic anhydride, and we have calculated the relative free energies for the formation of the product by using the following equations, as illustrated in Figure 2:

The calculated free energies, illustrated in Table 6, suggest an exergonic process in all presented cases, and therefore, the formation of the desired homodimeric product would be possible. It could be seen that the calculated free energies for the formation of the product, in the presence of the methoxy-group, which has electron-donating properties, would stabilize the product the most in all presented cases, regardless of the position in the 3-acetylcoumarin ring (C-6, C-7 or C-8). A similar stabilization is observed in 6-Cl- and 7-Et_2_N-products, except for the biscoumarin having 7-Et_2_N-, calculated via Equation (2) and illustrated in Figure 2, where the geometry optimization of the final product fails. If we consider the presence of the NO_2_ group in positions C-6 and C-7, a decrease in the calculated free energies for the formation of the desired product, in all presented cases, was observed due to the electron-withdrawing effect of the nitro-group as a substituent.

It should be noted that the calculated free energies for the formation of the product vary significantly depending on the formal reaction by which the energies are calculated. This could be due to several reasons, including (1) the metal salts used for the calculations; (2) the presence of the metal salts coordinated towards the product; and (3) the initial compounds considered for the formation of the product. Nevertheless, the calculated free energies follow the same trend in all presented cases: The formation of the products is stabilized by the presence of an electron-donating group as a substituent in the coumarin ring, and the electron-withdrawing groups destabilize the formation of the product. The strongest influence on the energetics of the reactions could be observed for the 7-NO_2_-substituted product. In all cases, the formation of the 7-NO_2_- substituted product is less favorable by at least 50 kJ/mol.

In our previews investigations [28,29], we assumed that the most probable reaction path for the formation of homodimeric biscoumarins includes a radical initiation, and we proposed that activation of the process could also depend on the substituent’s electronic effects at the benzene ring, which could favor or destabilize the formed intermediates. Here, we extended our study and performed both experimental and quantum-chemical investigations for the formation of biscoumarins of various 3-acetylcoumarin derivatives that have a substituent in the sixth, seventh, or eighth position in the benzene moiety with electron-donating or electron-withdrawing effects. The obtained results show that the formation of the homodimeric products of the 3-acetylcoumarin derivatives could be observed if the substituent is situated in positions C-6 and C-8. Two parallel pathways for biscoumarin formation following radical and ionic intermediates have been investigated (Appendix A). In the ionic mechanism (Appendix A), the stability of the intermediates and the free energy for the formation of the biscoumarin products are not that sensitive to the type of used metal salt. We have focused on the radical reaction path due to the stronger exergonic effect for the biscoumarin formation and clear dependence on the type of the applied metal salt.

To further prove the formation of the homodimeric products via the radical mechanism, and eventually the formation of some intermediates, we have investigated a model system that contains two radical structures (unsubstituted 3-acetylcoumarin radical) and a molecule of the metal salt (ZnCl_2_). The two monomers were aligned in a similar manner as in the homodimeric molecule, but the initial distance between the C-4 and C-4′ atoms was 2.34 Å. Upon geometry optimization of the initial configuration, a spontaneous formation of the bond between the C-4 and C-4′ atoms was observed, and simultaneously, the two Cl atoms from the metal salt formed a bond with the Zn atoms coordinated towards the coumarin moiety—one for each Zn atom (Appendix A for the initial and final geometries can be seen in the Appendix A).

Focusing on the proposed radical reaction path, as mentioned earlier, substituents at the C-6 and C-8 positions do not influence significantly the formation of the homodimeric products experimentally, which might be due to the lack of conjugation of the substituents with the formed radicals in position C-4. This could be seen from the NBO analysis and the changes in the charge distribution on the C-4 atom where the new C-C bond is formed, SOMO energy change, and the calculated free energies for the formation of the proposed radicals. Experimentally, a product could not be formed if the reaction was conducted using 7-methoxy-3-acetylcoumarin, which, even though the quantum-chemical calculations show an exergonic process for the formation of the homodimeric product, the formation of the proposed 7-methoxy-3-acetylcoumarin radical is less favorable compared to the unsubstituted 3-acetylcoumarin. The reason for that is the destabilization of the formed radical due to the electron-donating properties of the -OMe group. Moreover, the calculated free energies for the formation of the radicals indicate more endergonic process in the presence of an electron-donating group in position C-7 by at least 16 kJ/mol (with -OMe) and increases up to 39 kJ/mol for the 7-Et_2_N derivative. The reactivity of the obtained radicals depends on the electronic properties of the substituent, as well as the substituent’s position in the coumarin ring.

### Structural Characterization of the Compounds

The structure of the obtained dimers was assigned using 1D and 2D NMR spectra. The analysis of the spectroscopic data showed that new 3,3′,4,4′-tetrahydro-3,3′-disubstituted-4,4′-biscoumarins with additional functional groups in the benzene nucleus of the coumarin fragments were obtained. The spectra are characterized by a small number of signals determining the presence of symmetry elements in the newly synthesized compounds. The biscoumarins are characterized as configurational rotamers. The two substituted 3,4-dihydrocoumarin rings are in a preferred *anti*-relationship along the C4-C4′ bond. The presence of four stereogenic centers in the dimeric structures suggests that the final products are mixtures of stereoisomers. However, substituted tetrahydrobiscoumarins have been isolated mainly as *meso*-isomers with additional enolization at the COCH_3_ group in the third position, Figure 3, as previously observed for the biscoumarin species [28].

The insolubility of the obtained compounds, which might be due to π stacking, prompted us to use various deuterated organic solvents, from which acetone-d6 and trifluoroacetic acid-d seemed to be appropriate. In the ^1^H NMR spectra of substances **2b**–**h** in deuterated acetone, Table 7, the signals for the H-4 and H-4′ protons are detected as singlets without measurable spin–spin constants, with a chemical shift between 3.978 to 4.299 ppm. The protons from the -OH group of the enol form of the acetyl group are characterized by high frequencies and chemical shifts of 13.150 to 13.413 ppm. In the ^13^C NMR spectra of the two dimers, the chemical shifts for all carbon atoms are with values analogous to those of the 3-acetylcoumarin dimer [28].

Analysis of the NMR spectral data for products **2b**–**f**,**2h** suggests the presence of two rotational stereoisomers, in which all protons are chemically equivalent, Table 7. The ^1^H NMR spectra displayed two equally intense lines for the methyl groups from the COCH_3_ group, and the H-4 and H-4′ protons indicated the presence of two isomers. The chemical shifts of the two methyl groups of the enolized acetyl group, as well as the methyl groups of the methoxy groups in the sixth and eighth positions, could suggest an *anti*-conformation at the C4-C4′ bond. The ratio is a little dependent on the solvent. It seems that there are two possible stereoisomers with *anti*-disposition of the coumarin fragments in the *meso*-isomer (C_2v_ symmetry) and *anti*-disposition in the racemic conformers, Figure 3.

The insolubility of the prepared biscoumarins gave us the possibility to investigate the compounds in a solid phase. The data from CPMAS NMR are presented in Appendix A and Appendix A. The spectra confirmed the structures of the biscoumarins with a small number of signals and the appearance of signals for two stereoisomers. The presence of two isomers was confirmed based on the frequency of the C-3 and C-4 atoms, falling in the range of 91.0–95.7 ppm and 40.1–46.3 ppm, respectively. The corresponding enol forms were registered from the chemical shifts of both the deshielded C-3 atoms and the C atoms from the CH_3_C(OH)= fragment. It should be noted that, only in the spectra of compound **2g**, one isomer was detected.

Furthermore, we found that biscoumarins were soluble in trifluoroacetic acid, which opened more options for subsequent NMR studies. The dimers were characterized in deuterated trifluoroacetic acid using 1D and 2D NMR techniques losing the information for the signals from the enol forms in the range of 10–12 ppm. The relative ratios of the isomers are as follows: **2b_A_**:**2b_B_** = 1:1.02, **2d_A_**:**2d_B_** = 1.32:1, **2e_A_**:**2e_B_** = 3.5:1, **2f_A_**:**2f_B_** = 2.25:1, and **2h_A_:2h_B_** = 1:1.3. The chemical shifts of the H-4 protons are in the range 3.834–4.688. The protons from the benzene nuclei are registered with the corresponding number of protons and multiplicity, with the frequencies influenced by the electronegativity of the substituents present in the respective structures. In the ^1^H spectrum of **2h** were found the deshielding effect of the C=C(OH)CH_3_ group on the protons from the newly incorporated aromatic ring and the shielding effect of the neighboring ring to the methyl groups from the C=C(OH)CH_3_.

More assignments were performed by using 2D NMR spectra—{^1^H,^13^C} HSQC and NOESY. The data can be seen in the Appendix A. Heteronuclear correlation techniques, such as HSQC, clearly show the cross-peaks for the interaction between directly bonded H-C, which were assigned. The enol structure was well-characterized by the chemical shifts of the two quaternary carbons and the carbon atom from the methyl group of C=C(OH)CH_3_ and the correlations of the H-4/H-4′ protons and C-4. Analysis of the NOESY spectrum of compound **2d** showed the cross-relaxations of H-4/H-4′ and the H-5/H-5′ protons, H-4/H-4′ and protons from the methyl group from =C(OH)CH_3_, and CH_3_O-groups with H-7/H-7′ protons.

The data from the ^13^C NMR spectra of compounds **2b**–**h** have also shown the presence of two isomers. The corresponding frequencies for C-3 were in the range 94.0–95.3 ppm and, for C-4, 47.6–48.3 ppm. The signal for the quaternary C atom from the =C(OH)CH_3_ fragment has the highest frequency and is registered in the 182.8–183.5 ppm range. In the aromatic part of the biscoumarins, changes in the chemical shift of the carbon nuclei are registered, which are related to the influence of the electron-withdrawing substituents.

## 3. Materials and Methods

Ultrasonic irradiation was performed in an ultrasonic cleaning unit Elmasonic P with a frequency of 37 and 80 kHz and heating (Elma Schmidbauer GmbH, Singen, Germany). Melting points were determined with a Kofler hot-stage apparatus (Reichert Technologies, New York, NY, USA) and were used without correction. The IR spectra were recorded with a Specord IR 71, IR 75 spectrophotometer (Carl Zeiss, 73447 Oberkochen, Germany). Liquid chromatography–mass spectrometry analysis (LC-HRAM) was carried out on Q Exactive^®^ hybrid quadrupole-Orbitrap^®^ mass spectrometer (ThermoScientific Co, Waltham, MA, USA) equipped with a HESI^®^ (heated electrospray ionization) module, TurboFlow^®^ Ultra High Performance Liquid Chromatography (UHPLC) system (ThermoScientific Co, Waltham, MA, USA), and an HTC PAL^®^ autosampler (CTC Analyt-ics, Zwingen, Switzerland). The chromatographic separations of the analyzed compounds were achieved on a Nucleoshell C18 (100 × 2.1 mm, 2.7 µm) analytical column (Macherey-Nagel, Düren, Germany) using gradient elution at a 300 µL/min flow rate. The used eluent systems were A—0.1% formic acid in water and B—0.1% formic acid in CH_3_CN. Full-scan mass spectra over the *m*/*z* range 100–600 were acquired in positive ion mode at resolution settings of 140,000. The used mass spectrometer operating parameters were spray voltage—4.0 kV; capillary temperature—320 °C; probe heater temperature—300 °C; sheath gas flow rate 40 units; auxiliary gas flow 12 units; sweep gas 2 units (units refer to arbitrary values set by the Q Exactive Tune software); and S-Lens RF level of 50.00. Nitrogen was used for sample nebulization and collision gas in the HCD cell. All derivatives were quantified using 5 ppm mass tolerance filters to their theoretically calculated *m*/*z* values. Data acquisition and processing were carried out with the XCalibur^®^ ver 2.4 software package (ThermoScientific Co, Waltham, MA, USA).

^1^H and ^13^C NMR spectra were recorded on a Bruker Avance III 500 (Bruker BioSpin GmbH, Rheinstetten, Germany), at 500 MHz for ^1^H, 125.7 MHz for ^13^C, and Bruker Avance II+ 600 (Bruker BioSpin GmbH, Rheinstetten, Germany), at 600 MHz for ^1^H and 150.9 MHz for ^13^C, spectrometers (NMR Laboratory, Sofia University, Faculty of Chemistry and Pharmacy and NMR Laboratory—IOCCP, BAS). Chemical shifts are given in ppm from the solvent signal of deuterated trifluoroacetic acid at 11.5 ppm for ^1^H and 164.2 ppm for ^13^C; spectra. Chemical shifts are given in ppm using tetramethylsilane as an internal standard in acetone-d6. ^13^C CP MAS spectra were recorded at ambient temperature operated at 15 kHz spinning speed.

Reactions were monitored by TLC (Merck, Darmstadt, Germany) on silica gel 60 F_254_.

The substituted 3-acetylcoumarins were synthesized according to the procedures described in the literature [33,34].

### 3.1. General Procedure for the Preparation of Substituted 3,3′-Diacetyl-[4,4′-bichroman]-2,2′-dione

**For Method A** [28], a mixture of substituted 3-acetylcoumarin (0.001 mol), Zn (0.366 g, 0.0056 mol), (ClCH_2_CO)_2_O (0.410 g, 0.0024 mol) in Et_2_O/THF (10 mL/7 mL), and a catalytic amount of I_2_ was sonicated (water bath—40 °C) until the starting coumarin was consumed (TLC monitoring), Table 1. The reaction mixture was poured onto a 2N solution of hydrochloric acid and ice and extracted with chloroform (3 × 20 mL), and the organic extracts were washed several times with a saturated solution of NaHCO_3_ and dried with anhydrous sodium sulfate. The solvent was evaporated, and 3 mL Et_2_O and 1 mL acetone were added to the residue. The resulting mixture was left overnight in a fridge. Compounds **2b**–**h** were obtained as solids. After the filtration of the crystals, the solvent of the mother liquor was evaporated, and additional crystals were obtained. The combined solids were further purified by recrystallization from chlorobenzene.

**For Method B** [29], a mixture of substituted 3-acetylcoumarin (0.001 mol), Zn (0.262 g, 0.004 mol), Zn(CH_3_COO)_2_ × 2H_2_O (0.659 g, 0.003 mol), and a catalytic amount of I_2_ in THF (10 mL) was sonicated (water bath—–40 °C) until the starting coumarin was consumed (TLC-monitoring), Table 1. The reaction mixture was poured onto a 10 mL concentrated hydrochloric acid and extracted with dichloromethane (5 × 10 mL), and the organic layers were washed with water (2 × 15 mL) and a saturated solution of NaCl (2 × 15 mL). The extract was dried with anhydrous sodium sulfate, and the solvent was evaporated. Et_2_O (3 mL) and acetone (1 mL) were added to the residue, and the resulting mixture was left in a fridge overnight. Compounds **2d**–**g** were obtained as a solid. After the filtration of the crystals, the solvent of the mother liquor was evaporated, and an additional crystalline mass was obtained. The combined solids were further purified by recrystallization from chlorobenzene.

*3,3′-Diacetyl-[4,4′-bichroman]-2,2′-dione **2a***—IR (nujol): ν = 1595, 1645, 1450 cm^−1^ [28].

*3,3′-Diacetyl-6,6′-dimethoxy-[4,4′-bichroman]-2,2′-dione, **2b***—Method A. The product was isolated as white crystals from Et_2_O/acetone: 0.139 g, 64%, white crystals, m.p. = 196–198 °C, 202–206 °C (Et_2_O/acetone) and 187–192 °C (chlorobenzene). IR (nujol): ν = 1650, 1490, 1190, 1020 cm^−1^.

**2b_A_** (*minor isomer*): **^1^H NMR** (600 MHz, (CD_3_)_2_CO) δ = 13.244 (d, J = 0.6 Hz, 2H, =C(CH_3_)OH), 6.882–6.904 (m, 8H, aromatic), 6.465 (d, *J* = 1.9 Hz, 2H, H-5/H-5′), 4.023 (s, 2H, H-4/H-4′), 3.688 (s, 6H, two OCH_3_), 1.700 (s, 6H, two CH_3_).

**^1^H NMR** (500 MHz, TFA-d) δ = 7.316 (t, J = 8.0 Hz, 2H, H-8/H-8′), 7.196 (d, *J* = 8.2 Hz, 2H, H-7/H-7′), 6.618 (d, *J* = 7.4 Hz, 2H, H-5/H-5′), 4.053 (s, 6H, two OCH_3_), 3.964 (s, 2H, H-4/H-4′), 1.886 (s, 6H, two CH_3_). **^13^C NMR** (125.7 MHz, TFA-d) δ = 183.5 (s, enol form =C(CH_3_)OH), 173.76 (s, C-2/C-2′), 148.6 (s, C-8a/C-8a′), 142.5 (s, C-6/C-6′), 128.5 (s, C-7/C-7′), 124.04 (s, C-5/C-5′), 116.4 (s, C-8/C-8′), 125.3 (s, C-4a/C-4a′), 95.3 (s, C-3/C-3′), 48.3 (s, C-4/C-4′), 19.2 (s, CH_3_). **HRMS** (FTMS + p ESI) *m/z* calculated for C_24_H_22_O_8_ [M + H]^+^ 439.1387 found 439.1387.

**2b_B_** (*major isomer*): **^1^H NMR** (500 MHz, TFA-d) δ = 7.316 (t, *J* = 8.0 Hz, 2H, H-8/H-8′), 7.240 (d, *J* = 7.9 Hz, 2H, H-7/H-7′), 7.002 (d, *J* = 7.6 Hz, 2H, H-5/H-5′), 4.149 (s, 2H, H-4/H-4′), 4.080 (s, 6H, two OCH_3_), 1.990 (s, 6H, two CH_3_); **^13^C NMR** (150.9 MHz, CDCl_3_) δ = 183.3 (s, enol form =C(CH_3_)OH), 173.67 (s, C-2/C-2′), 148.4 (s, C-8a/C-8a′), 142.5 (s, C-6/C-6′), 127.9 (s, C-7/C-7′), 124.02 (s, C-5/C-5′), 116.2 (s, C-8/C-8′), 124.9 (s, C-4a/C-4a′), 94.5 (s, C-3/C-3′), 47.6 (s, C-4/C-4′), 19.1 (s, CH_3_). **HRMS** (FTMS + p ESI) *m/z* calculated for C_24_H_22_O_8_ [M + H]^+^ 439.1387 found 439.1387.

*3,3′-Diacetyl-8,8′-dimethoxy-[4,4′-bichroman]-2,2′-dione, **2d***—The product was isolated as white crystals from Et_2_O/acetone: Method A 0.199 g (91%), Method B 0.0916 g (42%); m.p. = 197–199.2 °C, 203–206 °C (Et_2_O/acetone) and 183–188 °C (chlorobenzene). IR (nujol): ν = 1650, 1640, 1600, 1590, 1490, 1470, 1190, 1080 cm^−1^.

**2d_A_** (*major isomer*): **^1^H NMR** (600 MHz, (CD_3_)_2_CO) δ = 13.150 (d, *J* = 0.6 Hz, 2H, =C(CH_3_)OH), 7.015–7.050 (m, 8H, aromatic), 6.440 (d, *J* = 1.9 Hz, 2H, H-5/H-5′), 3.978 (s, 2H, H-4/H-4′), 3.831 (s, 6H, two OCH_3_), 1.822 (s, 3H, CH_3_), 1.585 (s, 3H, CH_3_).

**^1^H NMR** (500 MHz, TFA-d) δ = 7.214 (t, *J* = 8.4 Hz, 2H, H-6/H-6′), 7.103 (d, *J* = 8.9 Hz, 2H, H-7/H-7′), 6.730 (d, *J* = 2.4 Hz, 2H, H-5/H-5′), 4.063 (s, 6H, two OCH_3_), 3.981 (s, 2H, H-4/H-4′), 2.005 (s, 6H, two CH_3_). **^13^C NMR** (125.7 MHz, TFA-d) δ = 183.3 (s, enol form =C(CH_3_)OH), 174.7 (s, C-2/C-2′), 158.1 (s, C-8a/C-8a′), 147.97 (s, C-8/C-8′), 131.5 (s, C-6/C-6′), 120.12 (s, C-7/C-7′), 125.3 (s, C-4a/C-4a′), 117.3 (s, C-5/C-5′), 95.2 (s, C-3/C-3′), 58.4 (s, OCH_3_), 48.2 (s, C-4/C-4′), 19.3 (s, CH_3_). **HRMS** (FTMS + p ESI) *m/z* calculated for C_24_H_22_O_8_ [M + H]^+^ 439.1387 found 439.1386.

**2d_B_** (*minor isomer*): **^1^H NMR** (500 MHz, TFA-d) δ = 7.214 (t, *J* = 8.4 Hz, 2H, H-8/H-8′), 7.197 (d, *J* = 8.9 Hz, 2H, H-7/H-7′), 6.948 (d, *J* = 2.8 Hz, 2H, H-5/H-5′), 4.085 (s, 6H, two OCH_3_), 4.138 (s, 2H, H-4/H-4′), 2.065 (s, 6H, two CH_3_). **^13^C NMR** (125.7 MHz, TFA-d) δ = 182.8 (s, enol form =C(CH_3_)OH), 174.5 (s, C-2/C-2′), 157.8 (s, C-8a/C-8a′), 147.92 (s, C-8/C-8′), 130.4 (s, C-6/C-6′), 120.09 (s, C-7/C-7′), 124.8 (s, C-4a/C-4a′), 117.2 (s, C-5/C-5′), 94.8 (s, C-3/C-3′), 58.3 (s, OCH_3_), 47.7 (s, C-4/C-4′), 19.2 (s, CH_3_). **HRMS** (FTMS + p ESI) *m/z* calculated for C_24_H_22_O_8_ [M + H]^+^ 439.1387 found 439.1386.

*3,3′-Diacetyl-6,6′-dibromo-[4,4′-bichroman]-2,2′-dione, **2e**—*The product was isolated as white crystals from Et_2_O/acetone: Method A 0.191 g (72%), Method B 0.170 g (64%); m.p. = 187–190 °C, 204–206.1 °C (Et_2_O/acetone) and 195–199 °C (chlorobenzene). IR (nujol): ν = 1650, 1595, 1590, 1240, 1180, 810, 520 cm^−1^.

**^1^H NMR** (600 MHz, (CD_3_)_2_CO) δ = 13.238 (d, *J* = 0.5 Hz, 2H, =C(CH_3_)OH), 7.574 (d, *J* = 2.4 Hz, 2H, H-5/H-5′), 7.533 (d, *J* = 8.6, 2.4 Hz, 2H, H-7/H-7′), 6.943 (d, *J* = 8.6 Hz, 2H, H-8/H-8′), 4.296 (s, 2H, H-4/H-4′), 1.750 (s, 6H, two CH_3_). **^13^C NMR** (150.9 MHz, (CD_3_)_2_CO) δ = 179.5 (s, =C(CH_3_)OH), 169.9 (s, C-2/C-2′), 151.3 (s, C-8a/C-8a′), 133.2 (s, C-7/C-7′), 132.8 (s, C-5/C-5′), 125.9 (s, C-4a/C-4a′), 119.2 (s, C-8/C-8′), 117.6 (s, C-6/C-6′), 92.8 (s, C-3/C-3′), 45.9 (s, C-4/C-4′), 18.2 (s, CH_3_).

**2e_A_** (*major isomer*) **^1^H NMR** (500 MHz, TFA-d) δ = 7.654 (dd, *J* = 8.7, 2.3 Hz, 2H, H-8/H-8′), 7.092 (d, *J* = 2.1 Hz, 2H, H-7/H-7′), 7.059 (d, *J* = 8.7 Hz, 2H, H-5/H-5′), 4.011 (s, 2H, H-4/H-4′), 2.169 (s, 6H, two CH_3_). **^13^C NMR** (125.7 MHz, TFA-d) δ = 183.5 (s, enol form =C(CH_3_)OH), 173.8 (s, C-2/C-2′), 158.1 (s, C-8a/C-8a′), 151.7 (s, C-8/C-8′), 134.9 (s, C-6/C-6′), 133.5 (s, C-7/C-7′), 125.2 (s, C-4a/C-4a′), 120.3 (s, C-5/C-5′), 94.5 (s, C-3/C-3′), 47.7 (s, C-4/C-4′), 19.4 (s, CH_3_). **HRMS** (FTMS + p ESI) *m/z* calculated for C_22_H_16_Br_2_O_6_ [M + H]^+^ 534.9386 found 534.9389.

**2e_B_** (*minor isomer*) **^1^H NMR** (500 MHz, TFA-d) δ = 7.642 (dd, *J* = 8.7, 2.3 Hz, 2H, H-8/H-8′), 7.410 (d, *J* = 2.1 Hz, 2H, H-7/H-7′), 7.020 (d, *J* = 8.7 Hz, 2H, H-5/H-5′), 4.099 (s, 2H, H-4/H-4′), 2.016 (s, 6H, two CH_3_). **^13^C NMR** (125.7 MHz, TFA-d) δ = 183.3 (s, enol form =C(CH_3_)OH), 174.1 (s, C-2/C-2′), 158.1 (s, C-8a/C-8a′), 151.6 (s, C-8/C-8′), 135.0 (s, C-6/C-6′), 133.8 (s, C-7/C-7′), 125.3 (s, C-4a/C-4a′), 120.5 (s, C-5/C-5′), 94.2 (s, C-3/C-3′), 48.1 (s, C-4/C-4′), 19.2 (s, CH_3_). **HRMS** (FTMS + p ESI) *m/z* calculated for C_22_H_16_Br_2_O_6_ [M + H]^+^ 534.9386 found 534.9390.

*3,3′-Diacetyl-6,6′-dichloro-[4,4′-bichroman]-2,2′-dione,****2f***—The product was isolated as white crystals from Et_2_O/acetone: Method A 0.157 g (71%), Method B 0.121 g (55%); m.p. = 190.4–193 °C, 204–206.4 °C (Et_2_O/acetone) and 184–196 °C (chlorobenzene). IR (nujol): ν = 1660, 1595, 1240, 1180, 810 cm^−1^.

**^1^H NMR** (600 MHz, (CD_3_)_2_CO) δ = 13.249 (s, 2H, =C(CH_3_)OH), 7.442 (d, J = 2.5 Hz, 2H, H-5/H-5′), 7.396 (dd, J = 8.7, 2.6 Hz, 2H, H-7/H-7′), 6.996 (d, J = 8.6 Hz, 2H, H-8/H-8′), 4.299 (s, 2H, H-4/H-4′), 1.755 (s, 6H, two CH_3_). **^13^C NMR** (150.9 MHz, (CD_3_)_2_CO) δ = 179.5 (s, =C(CH_3_)OH), 170.0 (s, C-2/C-2′), 150.8 (s, C-8a/C-8a′), 130.2 (s, C-7/C-7′), 130.2 (s, C-5/C-5′), 125.5 (s, C-4a/C-4a′), 118.9 (s, C-8/C-8′), 129.9 (s, C-6/C-6′), 92.8 (s, C-3/C-3′), 45.9 (s, C-4/C-4′), 18.2 (s, CH_3_).

**2f_A_** (*major isomer*)^**1**^**H NMR** (500 MHz, TFA-d) δ = 7.340 (d, *J* = 2.1 Hz, 2H, H-7/H-7′), 7.131 (dd, *J* = 8.7, 2.3 Hz, 2H, H-8/H-8′), 6.931 (d, *J* = 8.7 Hz, 2H, H-5/H-5′), 3.961 (s, 2H, H-4/H-4′), 1.827 (s, 6H, two CH_3_). **^13^C NMR** (125.7 MHz, TFA-d) δ = 183.3 (s, enol form =C(CH_3_)OH), 174.2 (s, C-2/C-2′), 151.1 (s, C-8/C-8′), 133.9 (s, C-6/C-6′), 132.0 (s, C-8a/C-8a′), 130.8 (s, C-7/C-7′), 125.0 (s, C-4a/C-4a′), 120.2 (s, C-5/C-5′), 94.0 (s, C-3/C-3′), 48.2 (s, C-4/C-4′), 19.2 (s, CH_3_). **HRMS** (FTMS + p ESI) *m/z* calculated for C_22_H_16_Cl_2_O_6_ [M + H]^+^ 447.0397 found 447.0397.

**2f_B_** (*minor isomer*)^**1**^**H NMR** (500 MHz, TFA-d) δ = 7.331 (d, *J* = 2.1 Hz, 2H, H-7/H-7′), 6.957 (dd, *J* = 8.7, 2.3 Hz, 2H, H-8/H-8′), 6.783 (d, *J* = 8.7 Hz, 2H, H-5/H-5′), 3.857 (s, 2H, H-4/H-4′), 2.009 (s, 6H, two CH_3_). **^13^C NMR** (125.7 MHz, TFA-d) δ = 183.4 (s, enol form =C(CH_3_)OH), 173.9 (s, C-2/C-2′), 131.9 (s, C-8a/C-8a′), 151.2 (s, C-8/C-8′), 133.6 (s, C-6/C-6′), 130.5 (s, C-7/C-7′), 124.8 (s, C-4a/C-4a′), 120.0 (s, C-5/C-5′), 94.5 (s, C-3/C-3′), 47.7 (s, C-4/C-4′), 19.4 (s, CH_3_). **HRMS** (FTMS + p ESI) *m/z* calculated for C_22_H_16_Cl_2_O_6_ [M + H]^+^ 447.0397 found 447.0396.

*2,2′-Diacetyl-1,1′,2,2′-tetrahydro-3H,3′H-[1,1′-bibenzo[f]chromene]-3,3′-dione, **2g**—*The product was isolated as white crystals from Et_2_O/acetone: Method A 0.0521 g (22%), Method B 0.0881 g (37%); (Et_2_O/acetone). IR (nujol): ν = 1650, 1640, 1600, 1590, 1490, 1470, 1190, 1080, 810 cm^−1^.

**^1^H NMR** (500 MHz, TFA-d) δ = 8.492 (d, *J* = 8.5 Hz, 2H, H-9/H-9′), 8.028 (d, *J* = 8.1 Hz, 2H, H-5/H-5′), 8.009 (d, *J* = 7.9 Hz, 2H, H-8/H-8′), 7.915 (d, *J* = 7.6 Hz, 2H, H-7/H-7′), 7.678 (d, *J* = 7.6 Hz, 2H, H-4/H-4′), 7.312 (d, *J* = 8.9 Hz, 2H, H-6/H-6′), 4.946 (s, 2H, H-1/H-1‘), 0.986 (s, 6H, two CH_3_). **^13^C NMR** (125.7 MHz, TFA-d) δ = 183.9 (s, =C(CH_3_)OH), 174. 5 (s, C-3/C-3′), 151. 0 (s, C-4a/C-4a′), 134. 4 (s, C-4/C-4′), 132. 9 (s, C-9/C-9′), 132. 5 (s, C-9a/C-9a′), 131.7 (s, C-5/C-5′), 129.6 (s, C-5a/C-5a′), 128.2 (s, C-8/C-8′), 123.7 (s, C-1a/C-1a′), 123.0 (s, C-7/C-7′), 118. 6 (s, C-6/C-6′), 93.6 (s, C-3/C-3′), 42. 4 (s, C-4/C-4′), 18.3 (s, CH_3_). **HRMS** (FTMS + p ESI) *m/z* calculated for C_30_H_22_O_6_ [M + H]^+^ 479.1489 found 479.1488.

*3,3′-Diacetyl-7,7′-dimethyl-[4,4′-bichroman]-2,2′-dione, **2h**—*The product was isolated as white crystals from Et_2_O/acetone: Method A 0.166 g (82%). M.p. = 178.4–181 °C (chlorobenzene). IR (nujol): ν = 1650, 1600, 1595, 1590, 1230, 1130, 810 cm^−1^.

**2h_A_** (*minor isomer*) **^1^H NMR** (500 MHz, TFA-d) δ = 7.181 (d, *J* = 8.6 Hz, 2H, H-5/H-5′), 6.880 (bs, 2H, H-8/H-8′), 6.802 (d, *J* = 7.7 Hz, 2H, H-6/H-6′), 4.029 (s, 2H, H-4/H-4′), 2.422 (s, 6H, two 7-CH_3_), 1.684 (s, 6H, two CH_3_). **^13^C NMR** (125.7 MHz, TFA-d) δ = 182.8 (s, =C(CH_3_)OH), 174.9 (s, C-2/C-2′), 152.5 (s, C-8a/C-8a′), 143.2 (s, C-7/C-7′), 130.8 (s, C-5/C-5′), 120.8 (s, C-4a/C-4a′), 118.9 (s, C-8/C-8′), 129.1 (s, C-6/C-6′), 96.3 (s, C-3/C-3′), 48.3 (s, C-4/C-4′), 21.4 (s, 7-CH_3_), 18.9 (s, CH_3_). **HRMS** (FTMS + p ESI) *m/z* calculated for C_24_H_22_O_6_ [M + H]^+^ 407.1489 found 406.1485.

**2h_B_** (*major isomer*) **^1^H NMR** (500 MHz, TFA-d) δ = 7.146 (d, *J* = 7.8 Hz, 2H, H-5/H-5′), 7.072 (d, *J* = 7.7 Hz, 2H, H-6/H-6′), 6.939 (bs, 2H, H-8/H-8′), 3.716 (s, 2H, H-4/H-4′), 2.425 (s, 6H, two 7-CH_3_), 1.729 (s, 6H, two CH_3_). **^13^C NMR** (125.7 MHz, TFA-d) δ = 182.9 (s, =C(CH_3_)OH), 174.9 (s, C-2/C-2′), 152.5 (s, C-8a/C-8a′), 143.5 (s, C-7/C-7′), 128.4 (s, C-5/C-5′), 120.5 (s, C-4a/C-4a′), 119.1 (s, C-8/C-8′), 130.5 (s, C-6/C-6′), 94.7 (s, C-3/C-3′), 47.0 (s, C-4/C-4′), 21.42 (s, 7-CH_3_), 18.9 (s, CH_3_). **HRMS** (FTMS + p ESI) *m/z* calculated for C_24_H_22_O_6_ [M + H]^+^ 407.1489 found 406.1481.

### 3.2. Computational Details

The quantum-chemical modeling was performed with the density functional theory (DFT) [35,36,37,38] using the Gaussian16 suite of programs [39]. For the calculations, the hybrid B3LYP exchange-correlation functional was used [40,41,42,43], coupled with a 6-31++G** basis set and polarizable continuum (PCM) [44] model. The PCM model describes the solvent as a homogeneous dielectric medium that is polarized around the solvated molecule. The chosen solvent for the calculations is THF in order to have a better correlation with the experimental results.

## 4. Conclusions

The effect of different substituents in the coumarin ring of 3-acetylcoumarin was studied both experimentally and theoretically. Several 3-acetylcoumarin derivatives (substituted at C-6, C-7, and C-8) were tested in the optimized reaction conditions under ultrasound irradiation, and the biscoumarin species-bearing groups at positions C-6 and C-8 were isolated in good yields and further characterized. It should be noted that EDGs as the MeO- in position C-7 did not favor the formation of biscoumarin products. These results are in agreement with the performed DFT calculations for the formation of a 7-MeO acetyl coumarin radical. The calculated free energies indicate that the formation of the radical is more endergonic when EDG is present at C-7 in the benzopyran moiety.

The elucidation of the effect was further investigated by performing an NBO analysis both in the initial substituted coumarins and in the formed radicals. The changes in the charge distribution in the C-4 position, where the new C-C covalent bond is formed, were investigated. The most notable difference observed is in the 7-methoxy-, 7-diethylamino-, and 7-nitro- derivatives, where the difference in the C-4 charge is much more prominent compared to the other formed radicals. The electron-donating properties of the methoxy- and diethylamino- groups play a major role in the conjugation between the substituent at the C-7 and C-4 atom, indicated also by the SOMO energy and core level shifts for the C-4, therefore altering the reactivity of the coumarin system to participate in the homodimerization process.

The DFT calculations show that the presence of substituents at the C-6 and C-8 positions in the coumarin moiety would not affect significantly the formation of a radical that is in compliance with the experimentally obtained products. The synthesis of 3,3′-diacetyl-7,7′-dimethyl-[4,4′-bichromane]-2,2′-dione **2h** could be considered as proof for the radical mechanism, as the calculated free energy for the formation of the 7-methyl-3-acetylcoumarin radical does not differ significantly from the unsubstituted 3-acetylcoumarin **1a**. Moreover, the difference in the NBO charges at the C-4 atom after the formation of the radical slightly alters in the presence of the 7-Me group in comparison with the 3-acetylcoumarin radical. Therefore, the experimentally observed effect of the substituents on the outcome of the reaction is in agreement with the theoretical calculations.

## Data Availability

Data are contained within the article and Appendix A.

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
