# Peer review of "Elucidating the Mechanism of Coumarin Homodimerization Using 3-Acetylcoumarin Derivatives"

_molecules, 2025, doi:10.3390/molecules30030651_

Round 1
Reviewer 1 Report
Comments and Suggestions for Authors
The review report is in the attached file.

Author Response
Dear Editor,
We are writing in response to the remarks, questions and comments given by the Reviewers on our manuscript ID: 3403258 and title: “Elucidating the Mechanism of Coumarin Homodimerization Using 3-Acetylcoumarin Derivatives”.
The text of the manuscript was fully revised according to the reviewers' comments and suggestions. All the changes were highlighted by Track Changes function.
Here are our answers to the comments:
Reviewer 1: The study explores the homodimerization of 3-acetylcoumarin derivatives, examining how substituents at C-6, C-7, and C-8 influence radical formation and reaction outcomes. Ultrasound- assisted synthesis and DFT calculations showed that substituents at C-6 and C-8 have minimal impact, while those at C-7 significantly affect radical stability based on electronic properties. Electron-donating groups destabilized radicals, often preventing product formation, while electron- withdrawing groups enhanced stability. Structural characterization using NMR confirmed stereoisomers with anti-relationships along the C4-C4' bond. The findings highlight the role of substituent effects and positioning in determining reactivity and product formation.
We would like to thank the reviewer for the valuable comments and the time spent on the detailed analysis of our work.
Comments:
The manuscript is interesting but lacks important theoretical studies that help complement and support the mechanisms of experimental radical formation. After implementing the suggested corrections, the manuscript could be considered for publication.
Comment 1: Since the reaction mechanism involves radicals, it is useful to graph the spin density of the molecules. This visualization allows readers to identify the positions of the radicals, thereby improving their understanding of the process.
Response 1: We have graphed the spin densities as Figure 1 in the main text. Analisys of the spin density is given in a new paragraph from line 133 to line 143 as follows:
“………….This conclusion is supported by the plots of the spin densities of the investigated radicals of the 3-acetylcoumarin and its derivatives, using isovalue of 0.005. The presented spin densities additionally indicate that the unpaired electrons are located mainly on the Zn-atom, coordinated towards the O-atoms, from the lactone ring and on position C-4, C-6, and C-8, Figure 1. Moreover, in positions C-5 and C-7, there isn’t any spin density, which further proves the fact that substituents in those positions do not affect the reaction and the formation of the desired homodimeric product directly.”
Comment 2: In line 163, the authors mention that atomic charges are an index of local reactivity. However, to more accurately assess the molecule's local reactivity, it would be advisable for the authors to calculate the Fukui indices. This would allow for a more precise identification of the regions of the molecule with nucleophilic or electrophilic characteristics, which could be directly related to the atoms with a higher tendency to form radicals. Furthermore, this analysis could be complemented by comparing these results with the previous spin density graph, providing a more comprehensive view of the reactivity patterns.
Response 2: We have calculated the Fukui indices for the investigated molecules and added them to the main text from line 173 to line 193. The results are listed in Table 4.
“To further understand the local reactivities of the 3-acetylcoumarin and its derivatives we have calculated and analyzed the Fukui indices. To analyze the radical formation, we focus on the ƒ0 values, as it represents the average of ƒ+ (electrophilic susceptibility) and ƒ– (nucleophilic susceptibility). A higher ƒ0 value suggests that the corresponding molecular site is more prone to radical formation, whereas lower values indicate reduced radical reactivity. The key observations from the calculated ƒ0 values indicate: 1) 8-OMe (ƒ0=0.088) shows the strongest tendency for radical formation. The strong electron-donating OMe-group at this position destabilizes the radical, making the C-4 atom highly reactive; 2) Substituents like 7-Me (ƒ0=0.076) and 6-OMe (ƒ0=0.069) also enhance radical formation, although not as strongly as 8-OMe; 3) 6-Cl (ƒ0=0.064) slightly increases the radical susceptibility compared to the parent coumarin H- (ƒ0=0.056); 4) 6-NO2 (ƒ0=0.043) and 7-OMe (ƒ0=0.049) lower the tendency for radical formation at C-4. This is due to the electron-withdrawing effects of -NO2 and the stabilizing resonance effect of the OMe-group.”
Coment 3: In lines 253 to 263, the authors discuss the non-formation of the product from 7-methoxy-3- acetylcoumarin, attributing this phenomenon to the process being more endergonic. However, the occurrence of a reaction primarily depends on its activation energy rather than whether the process is endergonic or exergonic. To provide a more robust explanation of the phenomenon, the authors should perform a Reaction Coordinate Index (IRC) calculation for this specific reaction and determine the Gibbs free energy. This would allow them to calculate the activation energy and compare it with that of similar reactions, thus providing stronger data to support their analysis and offering a comprehensive view of the reaction's thermodynamics and kinetics.
Response 3: We have performed quantum chemical calculations (for the limited time, that we have to prepare the answers) for the formation of the homodimeric product of 3-acetylcoumarin by investigating the ability to form the product via optimizing two 3-acetylcoumarin radicals in the presence of the metal salt (ZnCl2). It was observed that the formation of the product is a spontaneous process (barrier less at the given conditions), that occurs during the optimization of the system: The two C-4 atoms from the radicals form a bond and during this process, the coordinated Zn-atom to the O-atoms, from the acetyl group and the lactone ring, form a bond with the Cl-atoms from the molecule of the metal salt, resulting in the formation of the homodimeric product. This model system gives very useful information on the reaction: 1) it explains the necessity of a metal salt and metal Zn at the same time in the reaction conditions, something that is well known from the experiment and 2) the process is barrier less and the limiting step is the formation of the radicals. We have added a new paragraph in the main text from line 280 to line 289 and included figures for the initial and final geometries of the system in Figure S1, Supplementary materials.
We hope that we managed to answer the Reviewers’ comments and to make the appropriate changes to the manuscript.
Kind regards,
Prof. Petko Petkov, PhD
Prof. Rositca Nikolova, PhD
Reviewer 2 Report
Comments and Suggestions for Authors
Summary: In this paper, the authors describe a study on the reaction mechanism of 3-acetylcoumarin homodimerization using both experimental and theoretical manners. This study highlights the impact of substitutions at position C-7 on the stability of radical intermediates of coumarin derivatives, whereas substitutions at position C-6 and C-8 have negligible impacts regardless of the electronic properties. The reviewer thinks this paper fits into the scope of the journal and could be considered for publication after some minor comments are addressed.
Comments:
1. Line 52 and 58. Please add the degree sign before the C.
2. Line 127 and 173. Please define the EDG and EWG before presenting them in the manuscript.
3. Line 183 and 186. The value of the shift does not match with the mentioned entry, please correct this. For instance, 1.616 eV is the shift of entry 3, not 4.
I would like to talk a bit more about why I did not have many comments on this paper. From my perspective, the whole work was mostly conducted to explain the failure of a homodimerization reaction of a coumarin derivative. The authors hypothesized the differentiated impact of substitutions at different positions. This was later supported by DFT, NBO, and SOMO analysis, assuming the formation of the radical intermediates. The presentation of the data and the conclusions drawn from those data are straightforward and easy to comprehend. Therefore, I do not believe further comments are necessary.
Author Response
Dear Editor,
We are writing in response to the remarks, questions and comments given by the Reviewers on our manuscript ID: 3403258 and title: “Elucidating the Mechanism of Coumarin Homodimerization Using 3-Acetylcoumarin Derivatives”.
The text of the manuscript was fully revised according to the reviewers' comments and suggestions. All the changes were highlighted by Track Changes function.
Here are our answers to the comments:
Reviewer 2: In this paper, the authors describe a study on the reaction mechanism of 3-acetylcoumarin homodimerization using both experimental and theoretical manners. This study highlights the impact of substitutions at position C-7 on the stability of radical intermediates of coumarin derivatives, whereas substitutions at position C-6 and C-8 have negligible impacts regardless of the electronic properties. The reviewer thinks this paper fits into the scope of the journal and could be considered for publication after some minor comments are addressed.
We would like to thank the reviewer for the valuable comments and the positive evaluation of our manuscript.
Comments:
Comment 1: Line 52 and 58. Please add the degree sign before the C.
Response 1: We have added the degree signs in the corresponding lines.
Comment 2: Line 127 and 173. Please define the EDG and EWG before presenting them in the manuscript.
Response 2: We have added the definitions for the EDG (electron-donating groups) and EWG (electron-withdrawing groups) before presenting them in the text.
Comment 3: Line 183 and 186. The value of the shift does not match with the mentioned entry, please correct this. For instance, 1.616 eV is the shift of entry 3, not 4.
Response 3: We have corrected the values and the entry numbers in the text.
We hope that we managed to answer the Reviewers’ comments and to make the appropriate changes to the manuscript.
Kind regards,
Prof. Petko Petkov, PhD
Prof. Rositca Nikolova, PhD
Round 2
Reviewer 1 Report
Comments and Suggestions for Authors
No Comments